# Market Perspectives and Future Fields of Application of Odor Detection Biosensors within the Biological Transformation—A Systematic Analysis [note 1]

**DOI:** 10.3390/bios11030093

**Published:** 2021-03-23

**Authors:** Johannes Full, Yannick Baumgarten, Lukas Delbrück, Alexander Sauer, Robert Miehe

**Affiliations:** 1Fraunhofer Institute of Manufacturing Engineering and Automation IPA, 70569 Stuttgart, Germany; yannick.baumgarten@ipa.fraunhofer.de (Y.B.); lukas.delbrueck@ipa.fraunhofer.de (L.D.); alexander.sauer@ipa.fraunhofer.de (A.S.); robert.miehe@ipa.fraunhofer.de (R.M.); 2Institute for Energy Efficiency in Production (EEP), University of Stuttgart, 70569 Stuttgart, Germany

**Keywords:** odor sensor, market analysis, technology assessment, application field, performance profile, requirement profile, biointelligence, biological transformation

## Abstract

The technological advantages that biosensors have over conventional technical sensors for odor detection and the role they play in the biological transformation have not yet been comprehensively analyzed. However, this is necessary for assessing their suitability for specific fields of application as well as their improvement and development goals. An overview of biological basics of olfactory systems is given and different odor sensor technologies are described and classified in this paper. Specific market potentials of biosensors for odor detection are identified by applying a tailored methodology that enables the derivation and systematic comparison of both the performance profiles of biosensors as well as the requirement profiles for various application fields. Therefore, the fulfillment of defined requirements is evaluated for biosensors by means of 16 selected technical criteria in order to determine a specific performance profile. Further, a selection of application fields, namely healthcare, food industry, agriculture, cosmetics, safety applications, environmental monitoring for odor detection sensors is derived to compare the importance of the criteria for each of the fields, leading to market-specific requirement profiles. The analysis reveals that the requirement criteria considered to be the most important ones across all application fields are high specificity, high selectivity, high repeat accuracy, high resolution, high accuracy, and high sensitivity. All these criteria, except for the repeat accuracy, can potentially be better met by biosensors than by technical sensors, according to the results obtained. Therefore, biosensor technology in general has a high application potential for all the areas of application under consideration. Health and safety applications especially are considered to have high potential for biosensors due to their correspondence between requirement and performance profiles. Special attention is paid to new areas of application that require multi-sensing capability. Application scenarios for multi-sensing biosensors are therefore derived. Moreover, the role of biosensors within the biological transformation is discussed.

## 1. Introduction

This paper is the extended version of the proceedings paper presented at the 1st International Electronic Conference on Biosensors, 2–17 November 2020 [1].

The biological transformation is viewed by many as the next technological leap [2,3,4]. As such, it describes an interdisciplinary innovation pathway that seeks to increase the use of biological materials, structures, processes and organisms in technical systems through intensive collaboration and combination of production, information and biotechnology [2,5]. According to Miehe et al. [6], the development of the biological transformation can be divided into three modes. The first mode, bioinspiration, describes the transfer of evolutionarily optimized biological principles and processes to technology. In this context, the well-established research field of biomimicry (the transfer of phenomena, systems, elements or processes from nature to technology) is often mentioned, which has already led to numerous developments, for instance in lightweight construction or information technology (artificial intelligence, swarm intelligence, etc.). In addition to inspiration, first steps towards an integration of biological systems in production and products can be seen today. This second development mode includes the biotechnological production of vaccines or enzymes, for example as well as other innovative technologies, some of which show great potential, such as additive manufacturing of biobased materials [7]. The latter also include tissue engineering technologies that enable direct cell growth in such a way that specific shapes are achieved (e.g., bioprinting of functional tissues) [8,9]. The third development mode involves the interaction between technical, biological and information systems. This form of systemic interaction enables the creation of self-regulating value-added systems and products, which are summarized under the term biointelligence [6]. A system or product is considered to be biointelligent if there is a real time exchange of information between biological and technical components [10]. As part of a comprehensive pilot survey commissioned by the German government with regard to the biological transformation, biosensors were identified as one of the key enabling technologies. In this course, among others, odor-detecting biosensors were presented as promising examples [6,11].

Research has been conducted on odor sensors since the 1980s [12]. Over the years, various methods and technologies have been developed, ranging from analytical instruments such as mass spectrometry to smaller sensor arrays collectively known as “electronic noses”. Although analytical instruments are considered the “gold standard” in medical diagnostics, for example, the current trend is towards smaller and less expensive sensors for use in the field [13]. Numerous sensor arrays such as metal-oxide (MOX) sensors or carbon nanotubes [14,15], e.g., for the detection of chemical substances and gases (NO, O_2_, CO, H_2_, etc.) have demonstrated high sensitivity at low cost [13,16,17]. However, most of these sensors face challenges in terms of specificity, dependence on external conditions, and ability to discriminate between analytes, to name a few [13]. While advances are being made in electronic noses [18,19], biosensors appear to be a promising alternative. New developments in biotechnology, such as gene editing methods (CRISPR, etc.), make it possible to develop odor sensors that are able to identify gases and volatile organic compounds (VOCs) in low concentrations with high selectivity by specifically modifying the biological receptor components [20,21]. Because of their specific properties, these new technologies manage to open up new application possibilities. These include, for example, new forms of environmental monitoring or reliable testing of food quality [12]. It is essential to specify these application possibilities at an early stage in order to enable a more targeted market entry. To this date a comprehensive analysis of the requirements for application fields of odor detection sensors has not been presented nor has a meta-analysis showing the specific advantages of biosensors in this context. This paper thus addresses the following research questions:What are the specific market potentials of odor detection biosensors?What are therefore the most promising application fields for odor detection biosensors?What new fields of application can arise for odor detection biosensors from their specific properties?What role do odorant detection biosensors play in the biological transformation of industrial value creation?

This paper presents an evaluation method oriented towards the technology potential analysis of Spath et al. [22]. In this case, it is adapted to develop requirement profiles for different application areas and performance profiles for bio-based odor sensor technologies. By comparing the performance and requirement profiles, these questions can be answered and fundamental statements about application-specific market potentials for biosensors can be made. An overview of biological basics of olfactory systems is given in the following chapter and different odor sensor technologies are described and classified in order to create a common and consistent understanding.

## 2. Basics

All odor detecting biosensors are based on the fundamental principles of odor sensing in biological system. Therefore, the following section first gives an overview of the biological basics of the olfactory system before addressing the different odor sensor technologies and their classifications. Furthermore, the technical criteria used to create the performance profile are briefly described. At the end of this section, potential markets and applications for odor sensors are discussed.

### 2.1. Biological Foundations of the Olfactory System

Before addressing the available odor sensing technologies, the basic principles of odor sensing in biological systems are outlined. The system responsible for the reception of the stimulus initiated by odorants and the transmission and processing of the signals is referred to as the olfactory system. Stimulus perception occurs inside the nasal cavity on a sensory area of a few square centimeters, called the olfactory epithelium. Air reaches the olfactory epithelium via two access points, the orthonasal access during inhalation and the retronasal access during exhalation [23]. Figure 1 illustrates the structure of the olfactory epithelium which mainly comprises three cell types: olfactory sensory neurons, sustentacular cells, and basal cells. The olfactory sensory neurons are primary sensory cells and are constantly renewed as they lie unprotected on the surface and come into direct contact with hazardous substances in the breath [24]. They form dendrites, at the ends of which many small hairs, so-called cilia, are located which serve to enlarge the surface [25]. The cilia protrude into the mucus layer, which protects the skin from drying out and plays an important function in trapping VOCs [26]. The mucus contains water-soluble odorant-binding proteins that form a pocket for binding the VOCs and are therefore considered to play a major role in the transport the VOCs to the olfactory sensory neurons. These neurons contain receptors in the cell membrane of the cilia to which the VOCs bind and thus trigger transduction [23].

The transduction process of sensory olfactory neurons shown in Figure 1 begins upon binding of a VOC to a receptor. This G-protein-coupled receptor is a transmembrane protein in the olfactory cilia of the olfactory sensory cell. Extracellular binding of an odorant results in activation of the receptor, triggering an intracellular signaling cascade that is spatially separated from the receptor. The complex transduction process results in the influx of positive ions, namely calcium and sodium, and the efflux of chlorine ions.

After transduction, the intensity of the stimulus is mirrored in the amplitude of the triggered membrane potential. The potential is conducted to the axon hillock where, after reaching a threshold value, it is converted into electrical impulses of equal amplitude, called action potentials. In this way, the intensity of the stimulus is encoded in the frequency of the action potentials [27].

The process and structure for transmission and preprocessing of the action potentials before they enter the brain is substantially determined by the genetically encoded type of receptors. In the human genome, coding sequences for olfactory receptors are present in approximately 950–1000 genes, although more than 50% of these are nonfunctional due to mutations [25]. Expression of the active genes leads to the formation of a receptor. Only one type of receptor is formed per olfactory sensory neuron [23]. A receptor shows specificity for an odorant or a class of odorants of similar molecular structure. However, activation of the receptor is also possible by other odorants for which the receptor shows a lower affinity. At sufficiently high concentrations, the probability of binding substances for which the receptor shows a lower affinity increases, so that a signal may well result [25]. In addition, the receptor also determines where the signal from the olfactory sensory neuron is directed and processed. Thus, the axons of all olfactory sensory neurons of one receptor type lead to a collecting point, the so-called glomeruli, in the olfactory bulb. This is a significant first step of signal preprocessing since, therefore, subsequent processing structures only have to process signals from 4000 glomeruli instead of the signals from 100 million olfactory sensory neurons [26]. Accordingly, the principle that one odorant matches one specific receptor type, i.e., one olfactory sensory neuron, whose depolarization leads directly to odor sensation does not apply. Instead, several odorants can bind to several receptor types, i.e., various olfactory sensory neurons, to different degrees, which generates a characteristic pattern of activity in the olfactory epithelium that is transmitted via the olfactory signaling pathway to the brain for pattern analysis [23]. The ability to encode odorants combinatorically results in an almost unlimited differentiability of the very high variability of odorant molecules [25].

### 2.2. Types of Odor Sensing Technologies

Odor detection technologies can be divided into biosensors and technical sensors, each with several subgroups. Figure 2 illustrates a classification scheme developed in accordance with [20,28,29,30,31]. Biosensors contain integrated biological elements, such as cells, cell tissue, proteins, or nanovesicles, which are fundamental for their functionality and rely in part on the structures and processes outlined in the previous section. Technical sensors consist exclusively of technical components and can be divided into so-called electronic noses and conventional instrumental analysis. In the following, these technologies are briefly described.

An electronic nose is a technical system consisting of several chemical sensors that are connected to form a sensor array. Among the various sensor types shown in Figure 2, metal oxide (MOX) and conducting polymer sensors are the most commonly used [32]. These sensors form the detection element of the electronic nose and convert the chemical information into an analytical signal. Depending on the number and type of sensors used, the result of a measurement may be a complex signal pattern [33]. This pattern has to be compared with a reference pattern derived primarily from previous knowledge acquired from an existing data set. Only by matching the signal pattern with the reference pattern, a result with analytical significance is obtained [32]. Conventional analytical methods include, for example, gas chromatography and mass spectrometry. Here, the mixtures of substances in a liquid or gaseous state are examined for their chemical composition using different physical measuring principles, such as the detection of mass-to-charge ratio in mass spectrometry or polarity in gas chromatography [20]. Biosensors, rather than conventional sensors, use bio-elements such as proteins, nanovesicles, individual cells, or entire cell tissues as recognition elements. In a biosensor, the analyte to be measured docks to a bioreceptor. This creates a specific compound leading to a biochemical reaction that can be technically recorded and evaluated. For example, the reaction may involve a change in the thickness of the bioreceptor layer, the refractive index, light absorption, or electrical charge. These changes are detected by means of a transducer and converted into a signal, which is usually amplified and processed by an electronic system. Thus, a specific signal is generated for each specific substance [34].

As depicted in Figure 2, bioelectronic nose technologies comprise of three categories: cell-based, protein-based, and nanovesicle-based. Cave et al. [28] provides a comprehensive overview of these categories, which are briefly summarized below. A number of protein-based approaches aim to utilize olfactory receptor proteins embedded in membrane fragments, nanodiscs, or nanovesicles as the receiving element [35,36,37,38]. Membrane fragments of bacterial or yeast cells that express olfactory receptor proteins are solubilized with detergents in order to implement them in the sensor. The detergent acts as solubilizer and mimics the cell membrane environment, so that the olfactory receptor protein maintains its structure and function. This method is limited by missing other proteins, either in cytosol or membrane, relevant for e.g., signal transduction. Moreover, approaches in which odorant receptor proteins are isolated from cell membrane fragments and inserted into nanodiscs, synthetic membranes, including transmembrane proteins that help establishing an environment very similar to the original one, still separate the olfactory receptor proteins. Through sonication and centrifugation or chemical treatment of heterologous cells that express odorant receptor proteins along with proteins responsible for transduction, so-called nanovesicles can be formed. These are small spherical membrane fragments that contain all proteins for complete transduction cascade. Other authors see these approaches as a separate category of bioelectronic noses [29]. They may argue that spherical nanovesicles represent an intermediate form between cells and membrane fragments or nanodiscs. Irrespective of the applied introduced approaches, quartz microbalance electrode, electrochemical impedance spectroscopy, field effect transistors are often used as detectors. Challenges remain the intensive labor, reproducibility of performance, due to varying amounts of olfactory receptor proteins and variability in performance depending on temperature and humidity. Instead of monitoring the level of interaction of complex odorant receptor proteins with VOCs, simpler peptides derived from odorant receptor proteins can also be used offering the advantage of higher stability. Usually, these peptides are coupled with field-effect transistors and achieve a high sensitivity and reproducibility [39,40]. However, the number of available peptides is currently very limited [28]. As described in Section 2.1, odorant binding proteins are thought to facilitate the transport of VOCs through the mucus to the cilia of the olfactory sensory neuron. Due to this level of interaction with VOCs, they are interesting for an implementation in odorant sensors [41,42,43]. In comparison to sensors utilizing surface plasmon resonance, electrochemical impedance spectroscopy, sonic acoustic wave resonators, quartz microbalance electrodes those using field-effect transistors achieve a higher sensitivity. Advantages over odorant receptor protein include easier expression in heterologous cells and a higher stability towards environmental conditions. A major disadvantage lies in the limited amount of molecules being recognized due to very low specificity [28,44].

However, protein-based electronic noses have a limited lifetime, as their functionality decreases over time, due to the degradation and denaturation of receptor proteins and peptides. Another disadvantage is that necessary cofactors to restore the initial state after a transduction cascade must be actively provided from the outside, otherwise failure is imminent. Against this background, cell-based bioelectronic noses represent an attractive alternative through their ability to restore defective proteins and to produce the essential cofactors [28]. In approaches focusing on whole tissues, microelectrode sensors are inserted into the olfactory bulb of living animals to measure their neuronal activity when exposed to different odors [45,46]. Analyzing the activity patterns by computer programs to draw conclusions about specific odorants remains a complex challenge. Even approaches that involve existing olfactory epithelial tissue and placing it on microelectrode arrays, for example, encounter similar issues [47]. In addition, the preparation effort is very high and the comparability is low, since different signals are obtained depending on the recording area on the epithelium. The use of olfactory receptor organs from insects allow an easier analysis of distinct activity patterns, but the organs show stability issues [48,49]. Instead of using whole tissues, approaches using dissociated olfactory sensory neurons measure only single neurons that are either suspended, immobilized on a microelectrode array, or trapped in microfluidic chambers. The proof-of-concepts available so far, encounter the problem that there is no method to isolate or organize neurons of a specific receptor type, resulting in the readout signal being arbitrary [28,50,51,52].

Considering the advantages of cell-based sensor methods over peptide or protein-based ones as well as the outlined drawbacks of strategies utilizing whole tissues and dissociated olfactory sensory neurons, approaches using cultured cells are considered one of the most promising methods [11,28]. In cultured cells approaches, specific DNA segments encoding olfactory receptor proteins from olfactory neurons are transfected into a heterologous cell line so that it expresses olfactory receptors. Alternatively, there are strategies to extract olfactory receptor proteins and implant them into heterologous cells. The cells require a suitable culture environment (nutrient medium) on the sensor platform to remain functional and viable [28]. Veithen et al. [53] present a number of cell lines that are typically considered for in vitro expression and implantation of olfactory receptor proteins initially, regardless of the selection technology used. These include heterologous mammalian cells, such as the human embryonic kidney 293 cell line (HEK293) and Hela cells, as well as insect cells, Xenopus oocytes and yeast cultures [54,55,56,57,58,59,60]. To enhance the delivery of the expressed olfactory receptor molecule into the cell membrane, additional DNA segments of receptor transport proteins (RTP) are transcribed so that the cell line stably expresses chaperones such as the RTP1, RTP1s, RTP2, in addition to the expression of the olfactory receptor protein [56]. If the display of olfactory receptor proteins on the membrane succeeds, the binding of a VOC to this receptor triggers a reaction process that is comparable to the transduction cascade explained in Section 2.1 and also leads to the influx of calcium ions. One method to visualize this influx is by calcium imaging based functional assays. Calcium-sensitive indicators are used to detect deviations in the intracellular calcium concentration. For example, GCaMP is a genetically encoded calcium indicator that changes structure when calcium ions bind to it, activating its attached green fluorescent protein (GFP). Other commonly used calcium-sensitive dyes include Fura-2 and Fluo-4 [53,61]. Cell-based approaches are not limited to optical readout techniques, for instance, the surface plasmon resonance method or microelectrode arrays can also serve as transducers [28].

### 2.3. Technical Performance Criteria of Odor Sensing Technologies

In order to describe the performance of an odor sensor, both static parameters, such as selectivity and sensitivity, and dynamic parameters, such as service life, can be considered. In the following, the individual criteria considered in this paper are described. The selection and definitions were developed in the course of an expert workshop of the authors based on Fraden et al. and verified by the review and supplementation of external experts in different fields of sensor technology [62].

Measurement quality:

Sensitivity: describes the degree to which the output signal (measured value) changes in relation to the change of the input signal (measuring signal);Accuracy: describes the deviations of the sensor’s predicted measurement values from the real (ideal) value (typically 2 or 3 sigma of the error fluctuations);Selectivity: describes the response of the sensor to a certain group of analytes or one specific analyte;Specificity: indicates the probability that the measured value is falsely positive or falsely negative;Resolution: describes the smallest measurable change the sensor is able to register;Repeatability: indicates the error that occurs with repeated measurements, under the same situation.

Handling:

Reliability: describes the performance of the sensor that must be maintained over a defined period;Resistance to environmental influences or stability: describes the accuracy of the measurement results in case of changing environmental influences, such as temperature, humidity, radiation or magnetism;Maintenance effort: describes the overall effort of measures that keep the system in a functional state;Multi-sensing capability: describes the ability to measure several different substances in parallel;Operability: describes the simplicity of use;Measurement duration: the time required to complete a measurement process.

Technical construction and production:

Durability: describes the period of time during which the sensor remains functional, i.e., the performance remains within certain predefined specifications (e.g., a maximum drop of the measurable signal below 50% of its original value).Dimensions: describes the flexibility of relevant, characteristic geometric dimensions of the sensor shape.Weight: mass of the body in kg per measuring unit or sensor;Cost: describes the monetary costs of the manufacturing process for materials and the production process.

### 2.4. Markets and Application Fields for Biosensors

The annual turnover of all suppliers in the biosensor market was USD $11.5 billion in 2014 and is expected to grow to USD $28.78 billion by 2021. This corresponds to a growth rate of 12.2% per year [63]. In the following, the fields of application for the use of odor sensors are listed and described regarding their market volumes in Germany. Figure 3 gives an overview of the findings.

Healthcare: the healthcare market includes the ambulatory and stationary achievement contribution by established physicians, dentists, and hospitals, as well as other service providers [70]. In 2019, the health care system in Germany had a turnover of EUR 86.5 billion [64]. In 2018, 48,346 companies in Germany were active in the healthcare sector [71]. One example of a future field of application is diagnostics. Compared to healthy people, people with diseases excrete either different concentrations of certain VOCs or different types of VOCs. These VOCs are used as biomarkers and can be identified by breath, urine, and other body fluids. A diagnosis based solely on a patient’s odor requires very accurate diagnostic equipment [72]. Odor sensors prove to be a suitable diagnostic tool when it comes to diagnosing diseases. There is a great demand for non-invasive diagnostic methods in the healthcare sector. These sensor devices should be able to perform real-time monitoring, and they should be portable and inexpensive [73].Food industry: the food industry comprises food and feed manufacturers together with the beverage industry. Altogether, there are about 6000 companies with more than 20 employees in the German food industry [74]. In 2018, these companies employed more than half a million people. With an annual turnover of almost EUR 180 billion, the food industry is one of the largest industries in Germany [65]. The odor sensors in this industry should enable fast detection of quality changes during production. During quality control, impurities and pathogens are identified. Furthermore, the correct composition of the produced food and its smell and taste can be analyzed [75].Agriculture: agriculture is the economic activity where soil, livestock, labor, and know-how produce agricultural products that ensure the supply of plant and animal food to the people [76]. In 2018, there were 266,600 active companies in Germany [77]. They had a turnover of EUR 38.3 billion in 2018 [66]. Odor sensors can be used in agriculture to determine the quality of products and stocks based on odors or VOCs, or to detect pests and other negative influences already in the field [20]. Another application example is the monitoring of livestock odors [78].Cosmetics industry: cosmetics include all products that have a healing effect but are also used for beauty care. The industry is mainly determined by the large consumer goods groups. In 2018, there were 137 companies in the German industry for the production of cosmetics [79], generating sales of approximately EUR 6.4 billion [67]. Fields of application for odor sensors in the cosmetics industry are mainly quality control of production goods. Odor sensors can also be used in production to check the correct composition of the products, in order to be able to analyze odors and develop them more specifically, for example [20].Safety applications: safety applications are all applications that aim to detect hazardous substances. Smells contain important information about the environment and activities relevant to military and safety-oriented applications. This includes the detection of explosive materials or hazardous chemicals. However, an odor sensor can also be used for crime prevention tasks, such as security checks at airports or drug detection [80]. In 2021, the security industry in Germany is forecast to generate sales of EUR 9.2 billion [68].Environmental monitoring: in environmental monitoring, indoor and outdoor air is analyzed in order to detect air quality issues caused by harmful VOCs. These issues occur, for example, during the manufacturing of furniture [81]. The detection of harmful and toxic substances is also one of the areas of application for odor sensors. Furthermore, air quality and factory emissions can be monitored as well as the quality of ground and surface water. Sensors can be either installed stationary or mounted on drones [82]. Because of increased environmental awareness and pollution, the market for technological solutions for environmental monitoring applications is growing [75]. The turnover of the German environmental protection industry in 2018 amounted to EUR 71 billion [69].

## 3. Methodology

In each field of application (see Section 2.3) for odor detection sensors, there are different requirements for the technology used, which can be reflected in assessments of technical performance criteria. Sixteen technical performance criteria (see Section 2.2) that can be used especially for the description of the requirements of odor detection applications were established within this study through expert workshops and literature research. In order to specifically assess the importance of these criteria for the fields of application, a comprehensive expert survey was conducted with 11 experts from renowned research institutes and companies active in the fields of olfactory sensing electronic noses. Each of the participating experts had extensive experience in the research and development of odor sensors. The quality of the survey was, therefore, ensured by the targeted selection of experts who were able to classify the complex relationships between the product characteristics and their respective importance in the application fields and markets. The experts were asked to answer questions about which criteria were more or less important for each application field. For this purpose a scoring model was introduced to quantify the qualitative estimates for visualization, as follows: 0 = not important, 1 = rather unimportant, 2 = important, 3 = very important. In summary, the results of this survey were visualized in specific requirement profiles for each of the fields of application considered by forming the mean values of the scoring points. Additionally, all individual criteria were assigned to three related classes or categories. The first category combined all criteria related to measurement quality. This included the resolution and sensitivity of the sensors. The second category included the handling and the operability of the sensors. For example, measuring duration, maintenance effort, and multi-sensing-capability were assigned in this category. The third category combined production parameters such as manufacturing costs, weight, durability, and dimensions. Similarly, a performance profile was drawn up for individual technologies, showing the degree to which the performance criteria were fulfilled by the respective technology. The performance profile for bioelectronics noses was derived to enable statements about the fulfillments of the criteria in order to compare them to the competing technologies of technical sensor, as shown in Figure 2. To evaluate the performance of biosensors and instrumental analysis, numerous existing studies and research results were analyzed in a comprehensive meta-analysis regarding the performance perspectives of biosensors in comparison to those competing technologies. For the evaluation, the properties of biosensors were rated with the scale: 0 = is fulfilled worse by comparison; 1 = is fulfilled equally well or no clear statement can be made; 2 = is fulfilled comparatively better. In conclusion, by comparing the performance criteria with the requirement criteria, conclusions can be drawn about specific market potentials for the individual fields of application. In addition, an empirical investigation regarding the statistical variances of the survey results is analyzed for the assessments of importance for the respective criteria determined in the survey. The purpose of this investigation is to be able to conduct more in-depth research on criteria with high variances between the expert assessments in order to reduce the associated uncertainties. Criteria with high variances will consequently be examined and discussed in more detail to provide a better understanding of future market potentials. In particular, special attention will be paid to criteria that can potentially be better fulfilled by means of biosensors. This complementary analysis will serve as a basis for the goal of specific development targets for innovative biosensor concepts and business models. Furthermore, the analysis is supported by outlining the important role of biosensors against the background of the biological transformation.

## 4. Results

In the following subsections, the generated performance profile for biosensors (Section 4.1) and the requirement profiles of different application fields (Section 4.2) are presented. In addition, the market potential of biosensors is outlined for specific application scenarios within the application fields is outlined (Section 4.3).

### 4.1. Performance Profile of Biosensors for Odor Detection

The evaluation results show the performance of biosensors in comparison to technical sensors (electronic noses or instrumental analytics). All references and statements are summarized in Table 1 and the performance profile is graphically illustrated in Figure 4.

As illustrated in Figure 4, biosensors have advantages in terms of sensitivity, selectivity, specificity, accuracy, and resolution. This is due to physical bindings of the olfactory receptors with specific ligands. Therefore, the sensor can react to even very small amounts of analyte or single molecules within gas mixtures [83]. While there are advantages in terms of weight and dimensions compared to analytical instruments, this cannot be assumed for comparisons with electronic noses. The design of biosensors can be smaller than most technical analysis devices, such as mass spectrometers [75]. Disadvantages compared to technical sensors can be seen in terms of durability, maintenance effort, repeat accuracy, and resistance to environmental influences. The main reason for this is the limited lifetime and fragility of the used biomolecules. Users are forced to change the biomolecules after a certain time. This requires an enormous maintenance effort, which many users are not prepared to bear. Furthermore, the low resistance to environmental influences such as humidity, radioactive radiation, or high temperatures is a problem of biosensors that limits the application possibilities. All biosensors used, for example, for medical applications must meet the demanding and specific requirements of the medical industry. In addition, improvements and developments of other medical devices create more competitors for biological sensors [63]. A further disadvantage compared to technical sensors is cost. The long development cycles of biological sensors, which can only adapt to the new competitors with difficulty, play a key role here, according to the biosensor manufacturer Koniku Inc. Regarding the operability and the measuring duration, there are neither clearly defined advantages nor disadvantages for biosensors.

### 4.2. Requirement Profiles for Different Application Fields of Odor Sensing Technologies

In order to derive application-specific requirement profiles, each defined requirement criterion was evaluated with regard to its importance for a successful product in the respective fields of application. The evaluation was carried out in a survey, leading to the results shown in Figure 5, assigned to three related categories. The first category (a) combined all criteria related to measurement quality. The second category (b) included the handling and the operability of the sensors, and the third category (c) combined production parameters. The following sections describe the results grouped by these categories.

In Figure 5a, criteria are shown concerning the measuring quality of odor sensors. Overall, all the measurement quality criteria shown were assessed as “important” or “very important” across all application fields. In a comparison of the application fields, it can be seen that all quality criteria shown were of even higher importance for the application fields in the healthcare market and for safety applications, compared to the other fields. According to the experts, all quality-related criteria shown in Figure 5 were very important for these fields of application. Since critical safety and sometimes vital data are to be collected in these industries, high measurement quality is essential. For example, for the detection of explosives and medical diagnostics, it must be possible to detect even small trace elements and individual molecules with high specificity, sensitivity, accuracy, resolution, and selectivity. For safety applications, all criteria were rated 3, thus, as “very important.” The only exception for healthcare applications was high resolution, which was rated 2.75. High resolution was very important for all other fields of application with a rating of 2.75, as well, except agriculture. However, with a rating of 2.5, the criterion was still considered very important for agricultural applications.

In the category of handling and operation, shown in Figure 5b, there were stronger differences in the importance ratings for the considered fields of application compared to the criteria of measurement quality shown in Figure 5a. It is illustrated that short measuring times were very important for the food industry, due to the tendency of high throughputs of units to be measured coupled to large production numbers in this field of application. Because of the high risk of time delays, short measurement times were also very important for safety applications. According to the survey, the multi-sensing capability was particularly interesting and rated as important for the food industry, where taste analyses are performed. Tastes are usually defined by compositions of a large number of individual odorous substances. The multi-sensing capability was also evaluated as important for the cosmetics industry, since the composition of many different scents is also relevant for fragrances. The resistance to environmental influences was very important for applications in environmental monitoring, according to the experts, as these have to be used in changing environmental conditions outside the laboratory. This circumstance must not lead to any deviation of the measurement results. Resistance to environmental influences was also very important for safety applications and agriculture. In the cosmetics industry, however, this criterion was not very important, since the measuring systems can be used in a sterile and defined environment and fewer environmental influences are expected to affect the measurement results. Ease of operation or operability played an important role in all industries, since the measuring systems should be operable by ordinary employees who have no special training in the operation of these systems. For companies this was a decisive cost-saving factor, if no major training of the employees for the operation of the measuring system was necessary.

The criteria related to the construction and production of the sensors are summarized in Figure 5c. The geometric dimensions of the sensor tended to play a more important role in safety applications, since mobile applications, such as explosives’ detection or people searches are potentially more common there. This could also be the case for environmental monitoring, which is why the criterion for this field of application was also rated important. The weight of the sensors was also considered important for safety applications due to mobile applications. Rather unimportant ratings were, however, given to this criterion for environmental and agricultural applications. Weight tended to play a smaller role for mobile applications than dimensions. Due to the large areas to be monitored by sensors, drone applications can play a central role in agricultural applications in the future, which was the reason for the relatively higher importance of this field. Weight would be a decisive factor here. The durability rates varied in their importance for all application fields between a narrow range of 2 for the cosmetics industry and 2.5 for the environmental monitoring and food industry. Therefore, this criterion is important for all application fields. According to the experts, low cost production tended to play a more important role in the food and cosmetics industries than in the other fields of application rated as rather important. This could be due to the high competitive situation in this market, where manufacturing costs play a major role in gaining a competitive advantage over the competition.

The statistical variances of the survey results are summarized in Figure 6. It can be seen that in some cases there was a high degree of uncertainty regarding the assessment of the importance of technical performance criteria in certain fields of application. The measurement time in the healthcare market and the cosmetics industry as well as the multi-sensing capability for the health care market, safety applications, agriculture, and environmental monitoring should be emphasized, with variances higher than 2.

### 4.3. Market Potentials of Biosensors in Different Application Scenarios

To evaluate the findings, the experts in the Delphi survey were invited to assess the market potential of the applications known to date within the fields of application. In addition, the experts were asked in which future applications they see the use of odorant detecting biosensors. The economic potential of these new applications was also assessed in the second phase of the Delphi survey. The results are depicted in Figure 7. The economic potential was indicated on a scale from 1 = very low potential to 5 = very high potential.

The results of the requirement profiles of the application fields presented in Section 4.3 show that the requirements considered to be the most important were high specificity, high selectivity, high repeat accuracy, high resolution, high accuracy, and high sensitivity. These criteria describing the measurement quality were classified as “important” or “very important” in every considered field of application. All these criteria except for the repeat accuracy are potentially better met by biosensors than by technical sensors. It can be concluded that biosensor technology has a high potential for application in the considered fields and will play a decisive role in the market for odor sensors. Specific fields of application that can be covered specifically with biosensors, resulting from the high correspondence between requirement and performance profiles, are healthcare and security applications. These findings were confirmed by the results of the assessments of market potentials by the experts. In Figure 7a it can be seen that medicinal diagnostics, early cancer detection as well as detection of drugs or persons and detection of hazardous substances achieve high values from 4.0 and above.

Despite the high statistical deviations in multi-sensing capability, as shown in Figure 6, it can be considered as one of the key potentials, scoring 2.0 in importance among all application fields, as shown in Figure 5b. This finding can also be verified by the ranking of the newly identified application scenarios in Figure 7b. Food development, odor profiling, and ripeness assessment were ranked as having the highest market potential. All three application scenarios rely heavily on multi-sensing capability. Further elaboration and discussion on this topic is covered in the following chapter.

## 5. Discussion

In the following, the empirical investigation regarding the statistical variances of the survey results (Section 5.1) and the role of biosensors against the background of the biological transformation (Section 5.2) are discussed.

### 5.1. Discussion of the Empirical Investigation

An analysis of the empirical investigation regarding the statistical variances for the assessments of importance for respective criteria reveals a need for discussion of controversial statements and their importance in the markets under consideration. The experts agree on the importance of biosensors for the fields of application under consideration for most of the criteria that biosensors potentially fulfill better than conventional sensors, namely sensitivity, accuracy, resolution, selectivity and specificity (see Figure 5). However, the high variance within the expert responses regarding the multi-sensing capability indicates a high degree of uncertainty. Thus, it can be seen in Figure 6 that a comparatively high uncertainty across all fields of application occurred for the criterion of multi-sensing capability. Hence, a special focus is placed on the specific market potential associated with this criterion. To this end, specific application scenarios are described in the following for all fields of application that are directly related to the multi-sensing capability of the sensors. The identified and classified application scenarios provide approaches for specific development goals and are thus intended to reduce the identified uncertainties by providing ideas for specific application scenarios for multi-sensing capable sensors.

For healthcare market, technology leaps in diagnostics can be achieved. Diseases are often manifest through odorous substances secreted by the body before they can be detected with today’s analytical methods. Often, complex compositions of different molecules can be decisive in order to make a specific diagnosis. The fact that this is possible is known, for example, from studies in which dogs were trained to detect those substances. In this context the function that multi-sensing biosensors could fulfil can be seen. It was proven that through the odor substances, for example, tumor diseases in early stages but also mental disorders and even moods can be recognized purely on odor substances that are secreted by the body [65,66,67,68]. Imaging this capability with multi-sensing biosensors could thus represent a technological leap in medical diagnostics. Even after diagnosis, these sensors could be used to continuously monitor disease progression, providing a better basis for decisions regarding further treatment (see Figure 7b).

Tastes and smells play a major role in the food industry. In all cases, it is odor compositions and not individual analytes that play a decisive role. In order to be able to detect these compositions in a targeted and collected manner, multi-sensing sensors may play a decisive role in the future. Especially in quality assurance through the targeted in-line detection of fermentation or digestion processes, there may be an extremely high market potential for multi-sensing. As shown in Figure 7b, the development of new food products with predefined flavor profiles is another promising application enabled by multi-sensing. As in the food industry, odor compositions also play a decisive role in cosmetics industry. In addition to quality testing and assurance, multi-sensing can also provide a technological leap forward in research and development for odorants. The digital recording and visualization of odor profiles using multi-sensing biosensors for the targeted demand-oriented development of cosmetic products can become a game-changer in the cosmetics industry (see Figure 7b).

Further, the odor detection of safety applications can be continuously developed by multi-sensing. This means that a wide variety of hazards, such as pollutants or even traces of explosives or drugs in security-relevant areas such as airports, can be detected together in a single device. Clean air and water are very important for our health and key business cases for environmental monitoring. External influences, for example nitrogen oxide and particulate matter pollution in many cities with high traffic volumes, endanger it. But the health of our natural ecosystems is also affected by changes in the smallest particles in the air. To collect them in a multi-sensing device for the respective areas showing the indicators of health hazards or natural pollution as completely as possible would be a great step forward for environmental monitoring.

In addition, potentials through multi-sensing could be exploited in agricultural applications. Digitalization already plays a major role in the optimization of agricultural processes under the term “smart agriculture”. All related applications depend on suitable sensor technology. The integration of individual applications, such as pest detection, the degree of ripeness (see Figure 7b) and nutrition can be detected via various messengers. In order to be able to collect them, multi-sensing sensors could be used in the future. A more targeted nitrification, irrigation, and pest control and thus a reduction of the resources used and environmental impact can be achieved.

### 5.2. The Role of Odorant Sensing Biosensors within the Biological Transformation

As outlined in chapter 1 the biological transformation is progressing in three modes: bioinspiration, biointegration and biointelligence. The latter is characterized by the interaction between technical, biological and information systems. Consequently, a biointelligent system requires the implementation of an interface between biological and technical components. In addition to the identification of biosensors as key enabling technologies, so-called biology–technology interfaces (BTI) were identified as one of the core areas of future research [6]. The generic concept of a BTI comprises the recording and processing of information as well as control actions derived from it. The main interface components are corresponding sensors and actuators. They are based on electrical, chemical, mechanical or optical principles of action and realize a communication between the biological and technical system [10]. In this context, biosensors assume a special position as they can be considered an application of a BTI-based system on the one hand, and can also be seen as an enabler of superordinate BTI-based systems on the other. Odor detection biosensors provide a good example on both scenarios as illustrated in Figure 8. Firstly, the biological components (living cells, proteins, etc.) are in direct contact with the technical system (field-effect transistor, microelectrode array, etc.) and may be stimulated by adding VOCs and read out simultaneously. An information system evaluates the data and connects the biological and technical system, thus forming the basis of a BTI-based system. Secondly, the odorant detection biosensor may be deployed, for instance, in a food production bioreactor for an inline or online control. The reactor with its technical housing, its producing biological cells and the intelligent control system, based on the information of the odor detection biosensor along with other sensors form the superordinate BTI-based system. 

With regard to the categorization of odorant detection biosensors in the context of the three development modes of the biological transformation the state-of-the-art technologies described in Section 2.2 can be classified in the second mode, biointegration. While the integration of the biological components into the technical sensor is feasible, an intelligent readout and control system still leave room for improvement. Nonetheless, bioelectronic noses constitute the foundation of biointelligent systems (third mode of biological transformation), in particular due to their ability to map complex odor patterns (multi-sensing capability). As explained in Section 2.2, the evaluation of odor patterns is associated with intelligent information processing. Gutherie et al. [20] present various biomimetic approaches to the analysis of electronic sensor signals by using frameworks that mimic parts of the biological sense of smell (neural networks, etc.). One of the goals is to reproduce the temporally and spatially resolved information of the human sense of smell. Since these approaches are inspired by nature, they can in turn be assigned to the first mode of biological transformation, bioinspiration. However, as indicated earlier, they require further development to transform biosensors into fully biointelligent systems.

## 6. Summary and Outlook

In this paper, the specific market requirements for odor sensors were empirically assessed on the basis of 16 technical properties for various fields of application. The properties were classified into criteria concerning measurement quality, handling as well as construction and production-related criteria. The fulfillment of these criteria by biosensors compared to technical sensors was evaluated in order to derive specific market potentials. Biosensors were found to have advantages in measurement quality criteria (sensitivity, selectivity, specificity, accuracy, resolution), which are important for all application fields, especially for safety and healthcare applications. It can, therefore, be predicted that biosensors have comparatively high potential in these markets, with possible applications in the odor-based diagnosis and monitoring of various diseases or the detection of traces of drugs or explosives in security-relevant facilities. However, compared to technical sensors, disadvantages are seen in terms of durability, maintenance effort, repeat accuracy, cost, and resistance to environmental influences. Durability is rated as important to very important for all fields of application considered and should therefore be one of the focal points in the further development of biosensors. For applications in the cosmetics, food, and agricultural sectors, cost optimizations are necessary, since these markets are very price-sensitive due to either the high number of throughput and measurement cycles or high competition. For outdoor applications (environmental monitoring, safety, agriculture), resistance to environmental influences must also be improved. In addition, the analysis of the empirical investigation regarding the statistical variances of the survey results for the assessments of importance for the respective criteria determined in the survey has shown a high degree of uncertainty concerning the multi-sensing capability. Since this criterion also appeared to be an advantage of biosensors over technical sensors with moderate to high importance for all application fields, this uncertainty was addressed by identifying specific application scenarios in all application fields and providing approaches for specific development goals. Furthermore, in the context of the biological transformation, odorant detecting biosensors assume a special position as they are not only considered an application of a biology–technology interface (BTI) based system, but can also be seen as enablers of superordinate BTI-based systems (e.g., deployed in a bioreactor). However, for odorant detecting biosensors to make the step from biointegrated to truly biointelligent systems, further development in the area of pattern recognition (multi-sensing capability) is still necessary.

All in all, with the results obtained, market specific application potential and development goals can be discussed more clearly on the basis of qualitative assessments shown in this paper. However, the authors would like to point out that each application should be further regarded separately and can sometimes differ considerably from the requirement profiles of the respective application field. In addition, for investigations based on this results, weightings can be established for the criteria, the values of which can be determined from existing or future market volumes and the requirement profiles, for example. Furthermore, additional performance criteria, such as limit of detection, power consumption, and response time, as considered, e.g., by Burgués et al. [97,98], may be evaluated with respect to the performance profile of the biosensors as well as the application field requirements profiles. Moreover, there are strong dependencies between the specificity and selectivity criteria that make differentiation difficult and should therefore be discussed further. This paper can be referred to as a basis for further examinations.

## Figures and Tables

**Figure 1 biosensors-11-00093-f001:**
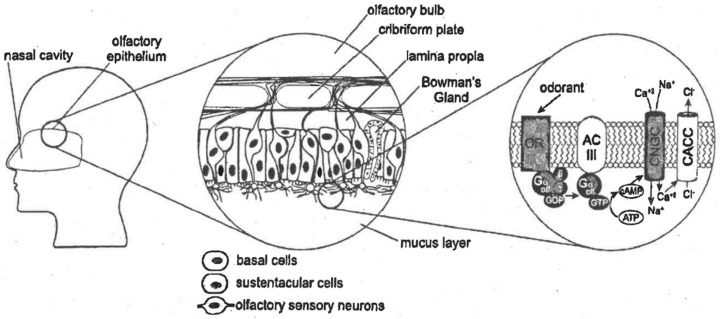
Structure of the human olfactory system according to Cave et al. [28]. The olfactory epithelium, located in the nasal cavity, mainly consists of basal and sustentacular cells as well as olfactory sensory neurons. On the upper side, neurons run through the cribriform plate of the skull towards the olfactory bulb of the brain. On the lower side, their thin cilia protrude in the mucus layer secreted by the Bowman’s gland. Odorant receptor proteins embedded in the ciliary membrane allow the binding of volatile organic compounds (VOCs) solubilized in mucus. Binding of a VOC to the receptor triggers a molecular transduction cascade, as shown on the right, resulting in an influx of mainly calcium ions. Additional efflux of chloride ions is thought to further increase the depolarization.

**Figure 2 biosensors-11-00093-f002:**
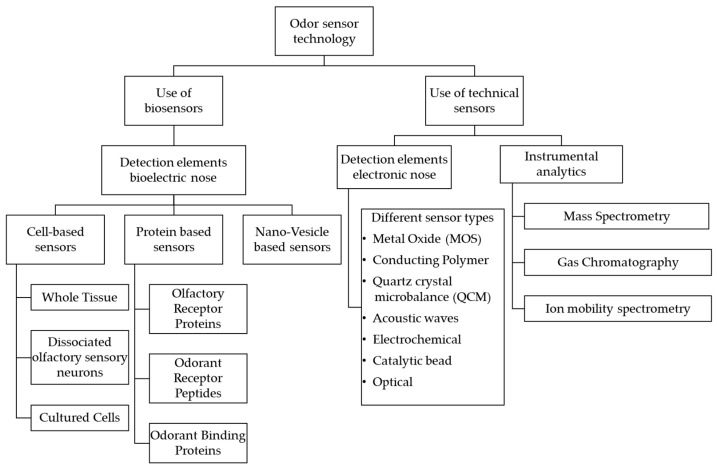
Overview of different technologies for odor detection. Own illustration, based on [20,28,29,30,31].

**Figure 3 biosensors-11-00093-f003:**
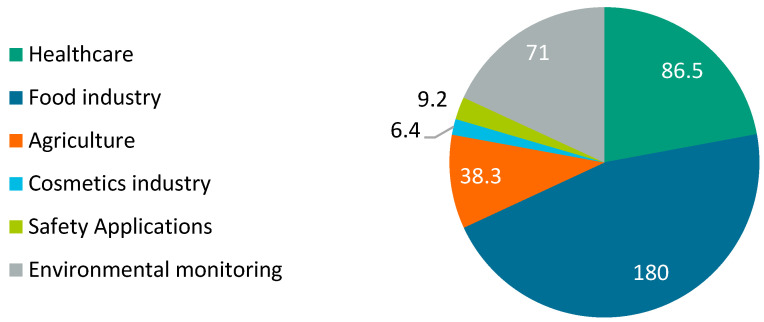
Overview of market volumes in billion euro (EUR) of different application fields in Germany [64,65,66,67,68,69].

**Figure 4 biosensors-11-00093-f004:**
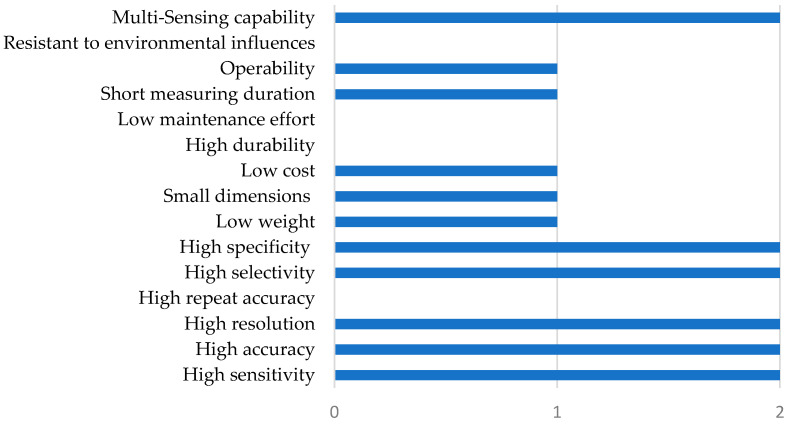
Performance profile of biosensors for odor detection in comparison to competitive technical odor sensors, based on Table 1; fields: 0 = is fulfilled worse by comparison; 1 = is fulfilled equally well or no clear statement to be made; 2 = is fulfilled comparatively better.

**Figure 5 biosensors-11-00093-f005:**
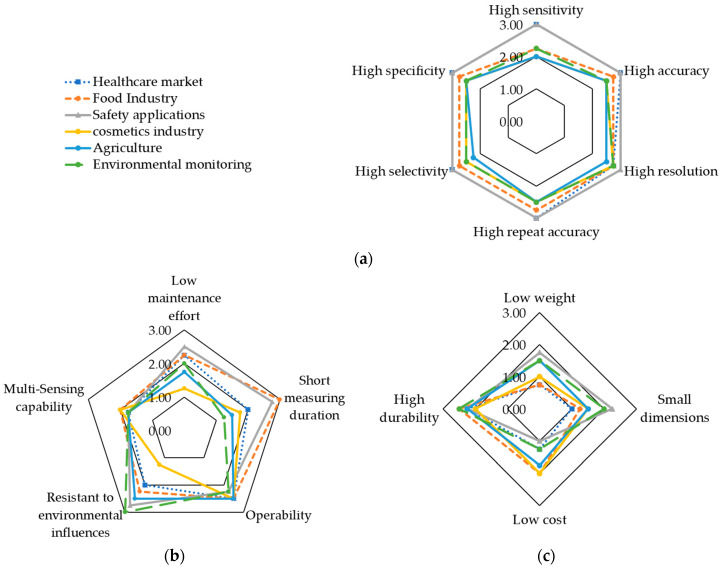
Evaluation of the requirement criteria on their importance for the categories of (**a**) measurement quality; (**b**) handling; (**c**) technical construction and production; 0 = not important, 1 = rather unimportant 2 = important, 3 = very important; sample size: 11.

**Figure 6 biosensors-11-00093-f006:**
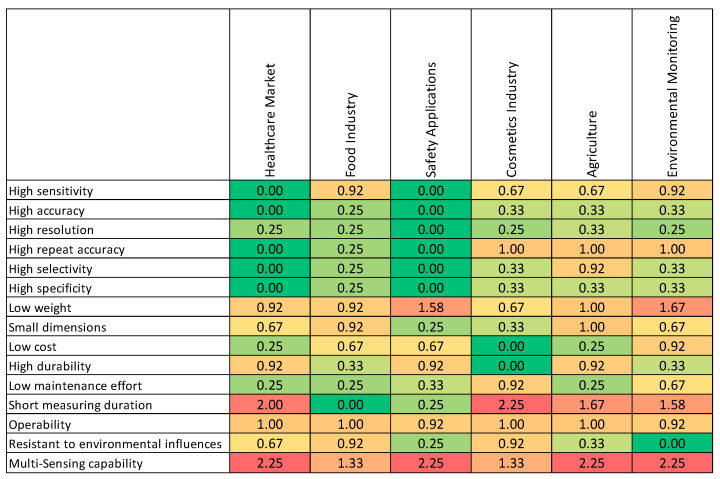
Statistical variances of the survey results shown in Figure 5 (red = higher values, green = lower values); sample size: 11.

**Figure 7 biosensors-11-00093-f007:**
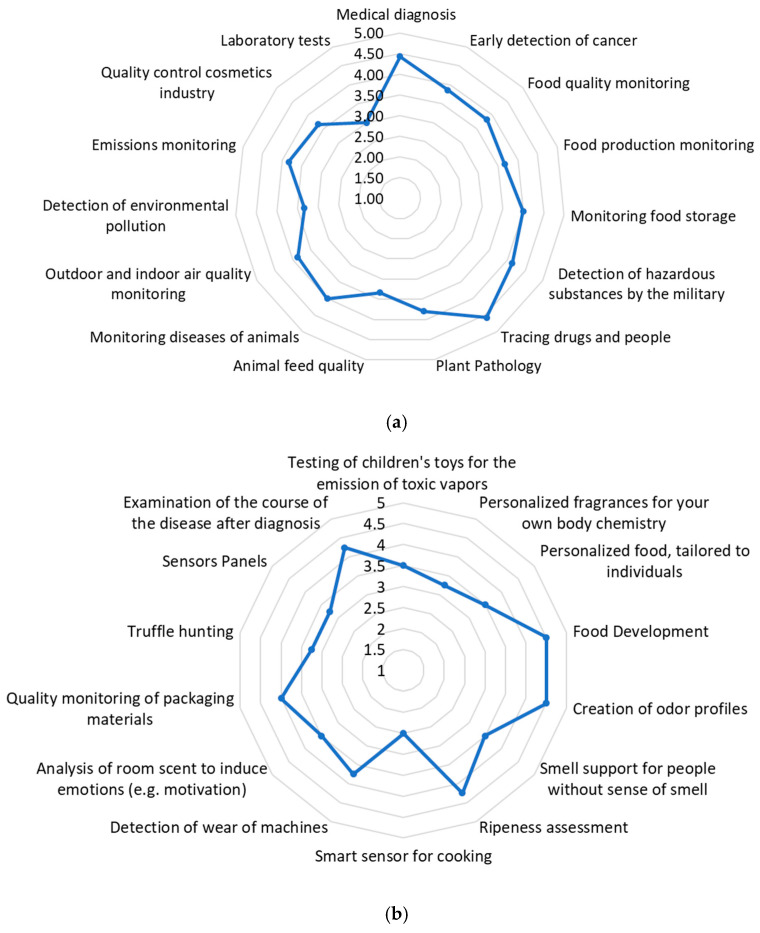
Results of the expert survey on the assessment of the market potential of (**a**) existing applications and (**b**) newly identified applications; 1 = very low potential, 2 = low potential, 3 = neutral potential, 4 = high potential, 5 = very high potential; sample size: 11.

**Figure 8 biosensors-11-00093-f008:**
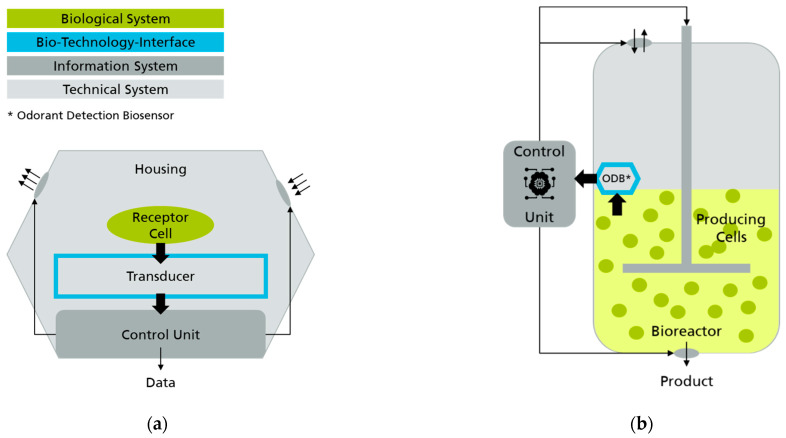
Odor detection biosensors as examples for biology–technology interface (BTI) systems following Miehe et al. [10]. Biosensors assume a special position as they can be considered as (**a**) an application of a BTI-based system, as well as (**b**) an enabler of a superordinate BTI-based system (i.e., bioreactor with cells that are generating a product).

**Table 1 biosensors-11-00093-t001:** Evaluation of the fulfillments of performance criteria by bioelectronic odor sensors; fields: 0 = is fulfilled worse by comparison; 1 = is fulfilled equally well or no clear statement to be made; 2 = is fulfilled comparatively better.

Properties	Fulfillment	Rating	References
High sensitivity	Because of the natural binding of olfactory receptors (ORs) with the specific ligand, the sensor can react even to very small amounts of analyte.	2	[12,83,84,85]
High accuracy	High accuracy due to natural binding of OR with specific ligand.	2	[12,83]
High resolution	Substances can be detected in very high resolutions at a level of nanomoles (or lower).	2	[12,86,87,88]
High repeat accuracy	Currently there are still problems with the stability of the results. No high repeat accuracy can be guaranteed yet.	0	[86,89,90]
High selectivity	It can be tested very specifically for certain substances.	2	[12,20,84,85]
High specificity	Good results for falsely positive and falsely negative measurements.	2	[12,20]
Low weight	A compact and light design for biosensors in comparison to analytical instruments allows online monitoring. Portable devices (sensors on chip) are currently in testing phases. No advantages. Probably no significant advantages over electronic noses to be expected.	1	[75,85]
Small dimensions	Analytical instruments are large benchtop systems permanently installed in laboratories. There are electronic noses with a diameter of a few cm. The same is possible for biosensors. Probably no significant advantages over electronic noses to be expected.	1	[17,85,89,91]
Low cost	The manufacturing costs for biological odor sensors are not yet finally known. Because of high research and development costs and complex production processes, a high sales price can be expected. For comparison, analytical instruments can cost up to USD 30,000. Electronic noses are available from USD 200.	1	[13,17,28,85,86,92,93]
High durability	Sensors, which use cells as bioreceptors, currently have a lifetime of just about a few weeks. The durability of these systems, especially for use as industrial sensors, are not reported.	0	[12,20,84,86,92]
Low maintenance effort	Bioreceptors must be replaced regularly. Replacement receptors must be stored correctly.	0	[28,86,94]
Short measuring duration	Measuring times for biosensors are reported from 5–30 s. Total measuring process takes 5 min due to sample preparation and pauses between measurements. This is comparatively faster than analytical instruments but in the same range as electronic noses.	1	[12,28,85]
Operability	Usability cannot be conclusively evaluated yet. However, odor sensors allow a non-invasive measuring method that does not require the extraction of sample material.	1	[33,73,85]
Resistant to environmental influences	Sensors must be protected against environmental influences. Susceptible to humidity and temperature fluctuations.	0	[20]
Multi-sensing capability	Biosensors are able to measure several different substances simultaneously. By multiplexing/multi-channeling various naturally occurring or synthetically optimized biological detection elements (olfactory receptors, olfactory receptor derived proteins, odorant binding protein), the bioelectronic nose can detect a variety of combinations of different VOCs. Although only a few multiplexed systems have been presented so far, multi-sensing is considered to be a decisive advantage over technical odor sensors in terms of mimicking and digitizing the sense of smell.	2	[29,33,83,84,85,95,96]

## Data Availability

Not applicable.

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
