# Peer review of "Market Perspectives and Future Fields of Application of Odor Detection Biosensors within the Biological Transformation—A Systematic Analysis†"

_biosensors, 2021, doi:10.3390/bios11030093_

Round 1

Reviewer 1 Report

This article by J. Full et al reports ‘Market Perspectives and Future Fields of Application of Odor Detection Biosensors within the Biological Transformation—A Systematic Analysis’. Authors studied specific market potentials of biosensors for odor detection by applying a tailored methodology that enables the derivation and systematic comparison of both the performance profiles of biosensors as well as the requirement profiles for various application fields. They assessed the fulfillment of defined requirements for biosensors using 16 selected technical criteria to evaluate a specific performance profile. Also, a selection of application fields for odor detection sensors was derived to compare the importance of the criteria for each of the fields, leading to market-specific requirement profiles. Authors reported that the requirement criteria include high specificity, high selectivity, high repeat accuracy, high resolution, high accuracy, and high sensitivity. In order to complete the literature reviews on these figures of merits, authors are recommended to enhance the introduction part using these recent and relevant papers;---Chemometrics heavy metal content clusters using electrochemical data of modified carbon paste electrode, Environ. Nanotechnol. Monit. Management, 14 (2020) 1003.  ---Curcumin-functionalized nanocomposite AgNPs/SDS/MWCNTs for electrocatalytic simultaneous determination of dopamine, uric acid, and guanine in co-existence of ascorbic acid by glassy carbon electrode. J Mater Sci: Mater Electron (2021). https://doi.org/10.1007/s10854-021-05282-1.   ---Mixed conductivities of A-site deficient Y, Cr-doubly doped SrTiO3 as novel dense diffusion barrier and temperature-independent limiting current oxygen sensors, Adv. Powder Technol. 31 (2020) 4657–4664.

Moreover, authors compared the performance of biosensors against technical sensors to derive specific market potentials or fields of application with higher potential for biosensors. They reported that compared to technical sensors, disadvantages are durability, maintenance effort, repeat accuracy, cost, and resistance to environmental influences. Authors further reported that cost optimizations are necessary for applications in the cosmetics, food, and agricultural sectors as these markets are price-sensitive due to either the high number of throughput and measurement cycles or high competition. Moreover, authors claimed that odorant detecting biosensors assume a special position in the context of the biological transformation because they are not only considered an application of a biology-technology interface (BTI) based system, but can also be seen as enablers of superordinate BTI-based systems (e.g. deployed in a bioreactor). Authors are recommended to use the following articles in which other applications of sensors and their importance, and performance are evaluated in details;---Diffusion kinetics mechanism of oxygen ion in dense diffusion barrier limiting current oxygen sensors, J. Alloys Compd. 855 (2021) 157465.  ---Mixed Conductivity Evaluation and Sensing Characteristics of Limiting Current Oxygen Sensors, Surf. Interfaces 21 (2020) 100762.  ---Mixed Conductivity and the Conduction Mechanism of the Orthorhombic CaZrO3 based Materials, Surfaces and Interfaces, Surf. Interfaces 23 (2021) 100905.  ---Ultra-High Selectivity of H2 over CO with a p-n Nanojunction based Gas Sensors and its mechanism, Sensor Actuat: B. Chem. 319 (2020) 128330.

Manuscript has however some grammatical mistakes which need to be improved. As a whole, the topic and presented results are interesting. Manuscript contains novelty and it is well organized and authors systematically addressed the corresponding issues. The conclusion is lengthy and needs to be shorten with the focus on the main points. I recommend publication of this article after major mandatory revisions and would like to see the revised version of paper before possible publication.

Author Response

Thank you very much for your revisions. Best regards

Reviewer 2 Report

The manuscript: "Market Perspectives and Future Fields of Application of Odor Detection Biosensors within the Biological Transformation - A Systematic Analysis",  presents an extended version of the proceedings paper presented at the 1st International Electronic Conference on Biosensors, 2-17 November 2020, Proceedings 2020, 60, 40, doi:10.3390/IECB2020-07029,  where most of the work is already published, to consider this new publication, the authors should rewrite the work adding new values from the study area, as it is now written, I do not consider it suitable for publication. The news regarding what is already published should be included in the abstract, in addition, line 31, "Therefore, biosensors thechnology in general has a high application potential for all the areas of application under consideration",the areas must be specified.

In Introduction:

1.biomimicry, biomechanics, bioprinting and biointelligent, all those terms must be defined.

2.pg 2 line 65, " different methods and technologies have been developed", an explanation must be added.

3. pg 2 line 68," new developments in biotechnology.....the authors must indicate the new developments in which they consist.

4. pg 2 line 85 " an evaluation method is presented", the authors must indicate that this new method is based.

5. Section 2 "Basics" must be included in the introduction, and this must be rewritten, making it clear from the beginning what is presented on this subject and what is new that this work wants to contribute.

6. Figure 2, Table 1, Figure 3, Figure 4 and Figure 5, They are exactly the same, that the authors have already published in the previous work in Proceedings.

7. pg 5 line  174-176, the definition of electronic nose is not correct, the whole paragraph must be modified.

8. pg 7 section 2.3, instead of listing all the criteria, they should group you according to some common category.

9. pg 7 section 2.4, This section should be summarized and include a pie chart of sectors, to give a quick and clear view of what is meant.

10. section 5 should be expanded and include applications that have already been developed with the exposed creatures, section 5.2 should expand its content.
The authors include a large number of bibliographic citations that are not commented on in the text.
For all these reasons, I think that it cannot be published in its current version, the authors would need to carry out an in-depth revision.

Author Response

(The authors gave the same response as above.)

Reviewer 3 Report

The authors present an interesting and comprehensive review of bioelectronic gas sensors, with clear introduction and methodology. However, despite they claim to have consulted a panel of experts to guide their research in the key figures of merits and applications of odor sensors, I feel that some concepts are not well defined and must be improved to ensure technical rigor. See my comments below:

L55-56: can you provide any reference/examples for biotechnological pharmaceuticals or bioprinting?

L63: typo: remove point before reference.

L65: "all these technologies.." -> can you provide some examples, e.g. MOX, electrochemical, etc.

L66: I don't think these technologies have low sensitivity. In particular, it is known that MOX and electrochemical sensors can detect ppb level concentrations. Some gravimetric sensors can detect mass changes of ng.

L67: Not clear what you mean with the breakthrough of odor sensors. Note that MOX sensors are used in CO alarms since the 1980s. Similarly, electrochemical sensors in breathalyzers for blood alochol content detection (BAC) etc... Just to mention few examples.

L70: Comparatively to what?

Fig. 2: Here you use "electric" nose while before you were using "electronic" nose. Ensure consistency. Electric is not the same as electronic. Electronic is a better suited term in this case.

L175: Not all chemical sensors are fabricated with CMOS technology. For example, amperometric gas sensors do not follow this fabrication principle.

L178: The output of a single sensor is not a complex pattern. It can be just an univariate signal. When you combine multiple sensors or operate them in complex ways (E.g. temperature modulation) you get complex patterns.

Section 2.3.
 - L290: Accuracy is not the maximum deviation, as there could be outliers. It is typically defined as 2sigma or 3sigma of the error (predicted-real) fluctuations.
 - Here you are missing an important one related to sensitivity: the limit of detection (LOD)->    https://www.sciencedirect.com/science/article/abs/pii/S0003267018301673
 - L294: Specificity is often the same as selectivity. What you are describing here is the specificity in terms of signal detection theory, which depends on the cut-off threshold applied to decide what is a "positive" and a "negative" in the particular application of binary classification problem.
 - L297: Repeatability: same measurement conditions better than same initial conditions.
 - L299: how to define functional? I would say that lifetime is the period when performance remains within certain predefined specs. In some cases it is defined when measurable signal falls below 50% of its original value under some conditions.
 - you are missing an important parameter: Power consumption -> https://www.mdpi.com/1424-8220/18/2/339
 - Another important missing parameter that you indeed use later: Response time.

 Section 2.4
  - L327: In many cases VOCs are the same but concentrations are different between healthy and ill subjects.
  - L349: Another important application: Monitoring of livestock odors -> https://link.springer.com/article/10.1007/s10661-007-9659-5
  - L365: The use of drones with electronic noses and chemical sensors for environmental monitoring is an important application -> https://www.sciencedirect.com/science/article/abs/pii/S004896972034701X

  Section 4.1.
   - Table 1: "There are electronic noses with a diameter of few cm" -> can you provide the reference/s?
   - Table 1: "Electronic noses are available from USD 200" -> can you provide the reference/s?
   - Table 1: High durability was previously referred to as lifetime. Ensure consistency.
   - Table 1: Multisensing capability. Why biosensors score 2? Analytical instruments and electronic noses can also several substances simultaneously...
   - L437: "there are advantages in terms of weight and dimensions" -> this contradicts the scores in the table for those parameters...

   Section 4.2
    - L475: the use of enoses for biomedicine is in practice for more than 20years -> https://www.degruyter.com/document/doi/10.1515/CCLM.2000.016/html
    - L514: "drone applications" -> this can be a relevant reference to include (https://www.sciencedirect.com/science/article/abs/pii/S004896972034701X)
    - Fig 4a: Why this one is so big compared to b and c)?

Author Response

(The authors gave the same response as above.)

Round 2

Reviewer 1 Report

Authors incorporated some of my comments and enhanced the quality of the paper, I recommend publication of this paper in the revised form.

Reviewer 2 Report

I consider that the changes included in the manuscript improve it considerably,, so I propose the manuscript for publication in Biosensors.

Reviewer 3 Report

I believe the authors have done an important effort to address the comments from all Reviewers. The new version has improved considerably from the original one. Congratulations.